# Optimal Kernel Choice for Score Function-based Causal Discovery

## Abstract

Score-based methods have demonstrated their effectiveness in discovering causal relationships by scoring different causal structures based on their goodness of fit to the data. Recently, Huang et al. (2018) proposed a generalized score function that can handle general data distributions and causal relationships by modeling the relations in reproducing kernel Hilbert space (RKHS). The selection of an appropriate kernel within this score function is crucial for accurately characterizing causal relationships and ensuring precise causal discovery. However, selecting the optimal kernel for a given data remains unsolved. In this paper, we propose a kernel selection method within the generalized score function that automatically selects the optimal kernel that best fits the data. Specifically, we model the generative process of the variables involved in each step of the causal graph search procedure as a mixture of independent noise variables. Based on this model, we derive an automatic kernel selection method by maximizing the marginal likelihood of the variables involved in each search step. We conduct experiments on both synthetic data and real-world benchmarks, and the results demonstrate that our proposed method outperforms heuristic kernel selection methods.

## 1 Introduction

Understanding causal structures is a fundamental scientific problem that has been extensively explored in various disciplines, such as social science (Antonakis & Lalive, 2011), biology (Londei et al., 2006), and economics (Moneta et al., 2013). While conducting randomized experiments are widely regarded as the most effective method to identify causal structures, their application is often constrained by ethical, technical, or cost-related reasons (Spirtes & Zhang, 2016). Consequently, there is a pressing need to develop causal discovery methods that can infer causal structures only from uncontrolled observational data. In recent years, score-based methods have emerged as a promising approach, enabling significant advancements in the field of causal discovery (Zhang et al., 2018; Vowels et al., 2022).

Generally, score-based methods are commonly employed for causal discovery by evaluating candidate causal graphs based on a specific criterion and selecting the graph with the optimal score (Chickering, 2002; Silander & Myllymäki, 2006; Yuan & Malone, 2013). Specifically, score-based methods begin by assuming a particular statistical model for the causal relationships and data distributions. Then, they proceed to test variables that are connected by the directed edges in the given graph, assessing their goodness to fit the data based on the assumed model and specific criterion. Subsequently, these methods score candidate graphs by accumulating all the local scores of the assumed edges in the given graph and select the graph with the optimal score. Most existing score-based methods only focus on one single specific model class, such as the BIC score (Schwarz, 1978), MDL score (Adriaans & van Benthem, 2008) and BGe score (Geiger & Heckerman, 1994) only for linear-Gaussian models, and BDeu score (Heckerman et al., 1995) which is specifically applicable to discrete data, as well as some other explicit model classes (Bühlmann et al., 2014; Hyvärinen & Smith, 2013; Sokolova et al., 2014) and their combinations (Claassen & Heskes, 2012; Tsamardinos et al., 2006).

Recently, Huang et al. (2018) proposed a generalized score function that can handle diverse causal relationships and data distributions. This approach utilizes the characterization of general independence relationships with conditional covariance operators (Fukumizu et al., 2004) to transform the

conditional independence test into a model selection problem. Therefore, it identifies the independence relationships among variables by modeling their causal relationships using a regression model in RKHS. Initially, a pre-defined kernel function is employed to map the dependent variables into an RKHS, which serves as the feature space. Following the regression model, these mapped features are assumed to be generated from their corresponding independent variables. Subsequently, the candidate graph is scored by estimating and accumulating the likelihood of the conditional distribution of each dependent variable with respect to its independent variables in the graph. To properly characterize the causal relationships, it requires to select an appropriate RKHS space as the feature space, which is achieved through a manually pre-defined kernel in this approach.

However, selecting the optimal kernel for given observational data is an important and challenging problem (Gretton et al., 2012). The default kernel parameters utilized in (Huang et al., 2018) are heuristically selected, which are boldly determined based on the distance between points in the original input space, without considering the specific characteristics of the data. Consequently, models employing such manually-selected and fixed kernel parameters may not accurately capture the true causal relationships, potentially leading to both spurious and missing edges in the causal graph, as will be demonstrated in Section 2.

**Contributions.** In this paper, we propose a novel kernel selection method to overcome the above limitation of the generalized score function with fixed kernels. Our proposed method can automatically select the optimal kernel parameters that best fit the given data. To achieve this, we extend the regression framework in RKHS and model the generative process of the involved variables in the graph as a mixture of independent noise variables. Correspondingly, we estimate the causal relationships of the involved variables by maximizing the marginal likelihood of their joint distributions instead of conditional distributions. In our approach, the selection of kernels is achieved through a learning process for the parameters of the kernels. That is, we select the kernel by optimizing its parameter with gradient-descent method, along with other parameters in the model simultaneously. Consequently, our method can automatically select the optimal kernel parameters that best fit the underlying causal relationship, which is superior to the manually-selected ones. We conduct experiments on both synthetic and real-world benchmark datasets. The experimental results demonstrate the effectiveness of our approach in selecting a better kernel while improving the accuracy of causal relationship discovery.

## 2 MOTIVATION

Huang et al. Huang et al. (2018) proposed a generalized score function that examines conditional independence relations by exploring the goodness to fit the data under a regression model in RKHS. Specifically, given a random variable $X$ and its independent variables $PA$ with domains $\mathcal{X}$ and $\mathcal{P}$, it assumes a causal relation in the RKHS, where the feature of $X$ in RKHS, denoted as $\boldsymbol{k}_x$, is generated through a nonlinear projection, represented as $f(PA)$, with the addition of independent noise $\epsilon$, mathematically,

$$\boldsymbol{k}_x = f(PA) + \epsilon, \tag{1}$$

where $\boldsymbol{k}_x$ is projected through a pre-defined feature mapping $\phi_{\mathcal{X}} : \mathcal{X} \to \mathcal{H}_{\mathcal{X}}$, which is related to a fixed kernel function $k_{\mathcal{X}}$. In the approach, the parameters within $f : \mathcal{P} \to \mathcal{H}_{\mathcal{X}}$ are learned by maximizing the likelihood of conditional distribution $p(\boldsymbol{k}_x | PA)$ to align with the data.

It can be observed from Eq.1 that the causal relationship between variables is characterized in the RKHS space with the pre-defined $\phi_{\mathcal{X}}$. Therefore the choice of kernel $k_{\mathcal{X}}$ related to $\phi_{\mathcal{X}}$, significantly influences the results of the independence test. Hence, the audacious use of heuristically-selected kernel parameters can lead to an unsuitable RHKS feature space, where the true causal relationships may not be accurately characterized. This can result in inaccurate assessments of the independence relationships among variables. Figure 1 depicts two failure cases using heuristically-selected kernel parameters. In both cases, we follow the default kernel setting as outlined in (Huang et al., 2018), using the Gaussian kernel and maintaining a fixed kernel bandwidth for $k_{\mathcal{X}}$ to twice the median distance between points in the input space.

In case 1, we consider a scenario where $X_2$ is generated from $X_1$ according to $X_1 = E_1$, $X_2 = \left(cos(X_1) + 0.5E_2\right)^2$ and the isolated node $X_3 = E_3$, with $E_1, E_3 \sim \mathcal{N}(0,1)$ and $E_2 \sim \mathcal{N}(0,0.5)$. Through the utilization of the predefined feature mapping $\phi_{\mathcal{X}}$ in conjunction with the fixed kernel function $k_{\mathcal{X}}$, the model will exhibit a spurious structure, $X_1 \to X_2 \leftarrow X_3$, where $X_3$ is considered

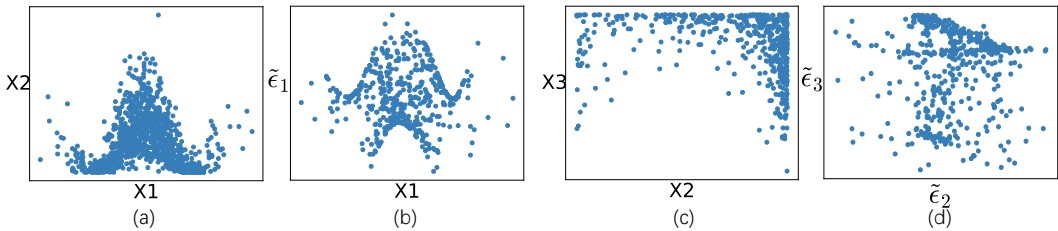

Figure 1: (a) Scatter plot of $X_1$ and $X_2$; (b) Scatter plot of $X_1$ and estimated noise $\tilde{\epsilon}_1$, which are not supposed to be correlated; (c) Scatter plot of $X_2$ and $X_3$; (d) Scatter plot of estimated noise $\tilde{\epsilon}_2$ and $\tilde{\epsilon}_3$, which are correlated.

as the parent of $X_2$. Figure 1(a) illustrates the scatter plot of $X_1$ and $X_2$, while Figure 1(b) displays the scatter plot of $X_1$ and the estimated noise $\tilde{\epsilon}_1 = \boldsymbol{k}_{X_2} - f(X_1)$. It is evident that the estimated $\tilde{\epsilon}_1$ exhibits a strong correlation with $X_1$. This phenomenon arises due to the inability of the model function, which is based on the unsuitable RKHS feature space determined by the manually-selected kernel parameters, to adequately capture the causal relationship, leading to an incorrect edge $X_3 \rightarrow X_2$ in the inferred graph.

In case 2, there are three variables $X_1, X_2$ and $X_3$, which satisfy $X_1 = E_1$, $X_2 = \cos(X_1^2 + E_2)$ and $X_3 = \cos(X_2^2 + E_3)$ with $E_1 \sim \mathcal{N}(0,1)$ and $E_2, E_3 \sim \mathcal{N}(0, 0.5)$. If we follow the default kernel setting in (Huang et al., 2018), an additional edge $X_1 \rightarrow X_3$ will be included in the inferred graph. Figure 1(c) presents the scatter plot of $X_2$ and $X_3$, while Figure 1(d) displays the scatter plot of the estimated noise terms $\tilde{\epsilon}_2$ and $\tilde{\epsilon}_3$, where $\tilde{\epsilon}_2 = \boldsymbol{k}_{X_2} - f(X_1)$ and $\tilde{\epsilon}_3 = \boldsymbol{k}_{X_3} - f(X_2)$. It can be observed that $\tilde{\epsilon}_2$ and $\tilde{\epsilon}_3$ exhibit correlation. This indicates that, with the pre-defined kernel $k_{\mathcal{X}}$, the model using the likelihood of the conditional distribution $p(\boldsymbol{k}_x|PA)$ fails to block the influence of $X_1$ on $X_3$, leading to the extra edge between $X_1$ and $X_3$.

In Figures 1(b) and 1(d), we only display the first dimension of the estimated noise $\tilde{\epsilon}$ in the feature space. Similar outcomes for the remaining dimensions are available in Appendix B.1. By demonstrating the potential limitations of the heuristically-selected kernels, we highlight the need for an automated method that can select appropriate kernel function, tailored to the specific data, to improve the accuracy of estimating causal relationships. In the following section, we will introduce our kernel selection method, which can automatically select the optimal parameters of $k_{\mathcal{X}}$ for the given data. With the kernel parameters selected through optimazation, our method is capable of inferring the correct causal structure for the two cases mentioned above.

## 3   Optimal Kernel Selection via Minimizing Mutual Information

In this section, we present a kernel selection method that enables the automatic selection of the optimal kernel for a given set of variables during the search procedure. To achieve this, we extend the RHKS regression to treat the originally-fixed kernel parameters to be trainable, enabling their optimization along with other parameters in the model. This allows us to automatically explore an appropriate feature space for accurate independent tests. Accordingly, we model the causal relationship as a mixture of independent noise variables in the RKHS and learn the parameters by estimating the maximum marginal likelihood of the joint distribution of the involved variables.

### 3.1   Preliminary

Before delving into the details of our method, we introduce the notations that will be used throughout the paper. Suppose there are three random variables $X$, $Y$ and $Z$ with domains $\mathcal{X}, \mathcal{Y}$ and $\mathcal{Z}$ respectively. For these variables, we have $n$ observations $D = \{(x_1, y_1, z_1), \cdots, (x_n, y_n, z_n)\}$. We define a RKHS $\mathcal{H}_{\mathcal{X}}$ on the domain $\mathcal{X}$, equipped with a continuous mapping function $\phi_{\mathcal{X}}$ : $\mathcal{X} \rightarrow \mathcal{H}_{\mathcal{X}}$ and a measurable positive-definite kernel function $k_{\mathcal{X}}(\cdot, \cdot) : \mathcal{X} \times \mathcal{X} \rightarrow \mathbb{R}$, which satisfies $k_{\mathcal{X}}(x_i, x_j) = \langle \phi_{\mathcal{X}}(x_i), \phi_{\mathcal{X}}(x_j) \rangle$. We denote $K_X$ as the kernel matrix of $X$ from $D$ with the element $K_{X(ij)} = k_{\mathcal{X}}(x_i, x_j)$. Therefore, for one particular observation $x \in \mathcal{X}$, we represent its empirical feature map as $\boldsymbol{k}_x = (k_{\mathcal{X}}(x_1, x), \cdots, k_{\mathcal{X}}(x_n, x))^T$. Similar notations are also applied to $Y$ and $Z$.

## 3.2 Regression in RKHS

Our proposed method is grounded in the RKHS regression framework in RKHS and leverages its inherent capability to characterize the independence relationship. Let us first introduce the connection between regression in RKHS and general conditional independence test. Suppose the random variables $X$, $Y$ and $Z$ satisfy $X \perp\!\!\!\perp Y \mid Z$. In the regression model in RKHS, the relationship is modeled as

$$\phi_{\mathcal{X}}(X) = f(Z) + \epsilon, \tag{2}$$

where $\phi_{\mathcal{X}} : \mathcal{X} \to \mathcal{H}_{\mathcal{X}}$, $f \colon \mathcal{Z} \to \mathcal{H}_{\mathcal{X}}$ and $\epsilon$ is the noise which is independent from $Z$. Let us denote $\ddot{Z} \coloneqq (Y, Z)$. According to the characterization of conditional independence with conditional covariance operators (Fukumizu et al., 2004), we have the following equivalence

$$E_Z[\mathrm{Var}_{X|Z}[g(X)|Z]] = E_{\ddot{Z}}[\mathrm{Var}_{X|\ddot{Z}}[g(X)|\ddot{Z}]] \iff X \perp\!\!\!\perp Y \mid Z \quad \text{for all } g \in \mathcal{H}_{\mathcal{X}}, \tag{3}$$

which implies that incorporating $Y$ as a predictor of $X$ given $Z$ is not advantageous, and as a result, we can confirm that $Y$ is not an independent variable of $X$. For the following two regression functions in RKHS

$$
\begin{aligned}
\phi_{\mathcal{X}}(X) &= f_1(Z) + \epsilon_1, \\
\phi_{\mathcal{X}}(X) &= f_2(\ddot{Z}) + \epsilon_2,
\end{aligned}
\tag{4}
$$

we can write $\phi_{\mathcal{X}}(X) = [\phi_1(X), \cdots, \phi_i(X), \cdots]^T$, with $\mathrm{Cov}\,(\phi_i(X), \phi_j(X)) = 0$ for any $i \neq j$. Since $\phi_{\mathcal{X}}(X)$ is the feature mapping in $\mathcal{H}_{\mathcal{X}}$, we can associate each component $\phi_i(X)$ with a function $g \in \mathcal{H}_{\mathcal{X}}$ such that $g = \phi_i(X)$, i.e. Eq. 3 holds for any $\phi_i(X)$. With the orthogonality between $\phi_i(Y)$ and $\phi_j(Y)$, we can derive that

$$E_Z[\mathrm{Var}_{X|Z}[\phi_{\mathcal{X}}(X)|Z]] = E_{\ddot{Z}}[\mathrm{Var}_{X|\ddot{Z}}[\phi_{\mathcal{X}}(X)|\ddot{Z}]] \iff X \perp\!\!\!\perp Y \mid Z. \tag{5}$$

Therefore, the assessment of conditional independence relations between the given variable and its potential independent variables in the graph can be transformed to evaluate the goodness to fit the data based on the regression model in RKHS. Huang et al. (2018) leverage this property and model causal relationship as a regression problem for the features mapped by the pre-defined $\phi_{\mathcal{X}}$. They further evaluate the model according to the maximum likelihood of the conditional likelihood $p(\boldsymbol{k}_x|Z)$ under the regression model. It is evident that the choice of kernel $k_{\mathcal{X}}$ plays a critical role in capturing the independent relationships within this model. However, the kernel parameters in (Huang et al., 2018) are manually selected and fixed. Unfortunately, the feature space corresponding to such a boldly-selected kernel may not consistently align with the underlying causal relationships, which has been demonstrated in the aforementioned cases.

Nevertheless, if we allow the kernel $k_{\mathcal{X}}$ to be trainable, it struggles to learn a non-trivial feature mapping that can capture causal relations with conditional likelihood-based score functions. This is primarily due to the absence of constraints imposed on $k_{\mathcal{X}}$. When learning the kernel parameters with the conditional likelihood-based score functions, the model will end up learning a trivial constant mapping with an excessively high score, which is uninformative for causal discovery purposes. Therefore, when utilizing a trainable kernel, it becomes imperative to employ a different score function that can impose constraints on the kernel parameters, ensuring the accurate capture of the true causal relationship.

## 3.3 Mutual information-based score function

In this section, we propose a novel score function that addresses the challenge of learning non-trivial relationships using trainable kernel parameters within the RKHS regression framework. We extend the regression model in Eq. 2 by incorporating a flexible function $\phi_{\mathcal{X}}(\cdot\,;\sigma_x)$, which is associated with a trainable kernel $k_{\mathcal{X}}(\cdot, \cdot\,;\sigma_x)$. Here, $\sigma_x$ represents the trainable parameters of the kernel, such as the bandwidth when $k_{\mathcal{X}}$ is a Gaussian kernel. Mathematically, we represent the extended model as

$$\phi_{\mathcal{X}}(X;\sigma_x) = f(PA;\theta) + \epsilon, \tag{6}$$

where $\phi_{\mathcal{X}} : \mathcal{X} \to \mathcal{H}_{\mathcal{X}}$, $PA$ denotes the independent variables of $X$, which may consist of multiple variables. Both $\sigma_x$ and $\theta$ are trainable parameters in our method.

Since $\phi_{\mathcal{X}}(X) \in \mathcal{H}_{\mathcal{X}}$ is in an infinite-dimensional space, where the probability is hard to measure, we practically map $\phi_{\mathcal{X}}(X)$ into its empirical feature $\boldsymbol{k}_x$. Suppose that we have a set of observations $D = \{(x_1, \boldsymbol{pa}_1), (x_2, \boldsymbol{pa}_2), \cdots, (x_n, \boldsymbol{pa}_n)\}$ for $(X, PA)$. With the property that functions in the RKHS are in the closure of linear combinations of the kernel at given points, mapping $\phi_{\mathcal{X}}(X)$ into $\boldsymbol{k}_x$ does not cause loss of information (Schölkopf et al., 2002). For one particular observation $(x, \boldsymbol{pa})$, the empirical feature of $x$ is represented as $\boldsymbol{k}_x = (k_{\mathcal{X}}(x_1, x; \sigma_x), k_{\mathcal{X}}(x_2, x; \sigma_x), \cdots, k_{\mathcal{X}}(x_n, x; \sigma_x))^T$. Therefore, the extended model on finite observations $D$ is reformulated as

$$\boldsymbol{k}_x = f(\boldsymbol{pa}\,; \theta) + \boldsymbol{\epsilon}, \tag{7}$$

where $\boldsymbol{\epsilon} = (\epsilon_1, \cdots, \epsilon_n)^T$ is the noise vector, with each element being independent of the others.

In our extended model defined in Eq. 7, the parameters $\sigma_x$ and $\theta$ are trained to make the estimated noise $\tilde{\boldsymbol{\epsilon}} = \boldsymbol{k}_x - f(PA; \theta)$ and $PA$ as independent as possible. This is achieved by minimizing their mutual information as following:

$$I(PA, \tilde{\boldsymbol{\epsilon}}) = H(PA) + H(\tilde{\boldsymbol{\epsilon}}) - H(PA, \tilde{\boldsymbol{\epsilon}}), \tag{8}$$

where $H(PA, \tilde{\boldsymbol{\epsilon}})$ is the joint entropy and $\tilde{\boldsymbol{\epsilon}} = (\tilde{\epsilon}_1, \cdots, \tilde{\epsilon}_n)^T$ is the estimated noise in the feature space. Based on the change-of-variables theorem, the joint density $p(PA, \tilde{\boldsymbol{\epsilon}})$ can be derived from $p(PA, \tilde{\boldsymbol{\epsilon}}) = p(PA, X)/|\det \mathbf{J}|$ and $\mathbf{J}$ is the Jacobian matrix of transformation from $(PA, X)$ to $(PA, \tilde{\boldsymbol{\epsilon}})$, i.e. $\mathbf{J} = [\partial(PA, \tilde{\boldsymbol{\epsilon}})/\partial(PA, X)]$. Therefore, the joint entropy of $(PA, \tilde{\boldsymbol{\epsilon}})$ is derived as

$$H(PA, \tilde{\boldsymbol{\epsilon}}) = -E[\log p(PA, \tilde{\boldsymbol{\epsilon}})] = -E\left\{\log p(PA, X) - \log|\mathbf{J}|\right\}. \tag{9}$$

By combining Eq.8 and Eq.9, we can express $I(PA, \tilde{\boldsymbol{\epsilon}})$ as follows:

$$\begin{aligned}
I(PA, \tilde{\boldsymbol{\epsilon}}) &= H(PA) + H(\tilde{\boldsymbol{\epsilon}}) - E[\log|\det \mathbf{J}|] + E[\log p(PA, X)] \\
&= H(PA) - E[\log p(\tilde{\boldsymbol{\epsilon}})] - E[\log|\det \mathbf{J}|] - H(PA, X) \\
&= -E[\log p(\tilde{\boldsymbol{\epsilon}})] - E[\log|\det \mathbf{J}|] + C,
\end{aligned} \tag{10}$$

where $C = -H(PA, X) + H(PA)$ does not depend on $\sigma_x$ and $\theta$, and thus can be considered as constant. Hence, we only focus on the first two terms in Eq. 10, which is equivalent to the negative log likelihood of $p(PA, X)$ when $\boldsymbol{\epsilon}$ is considered as Gaussian. Refer to Appendix A.1 for the detailed derivations. Therefore, for the particular observation $(x, \boldsymbol{pa})$, according to Eq. 10, we have the following negative log likelihood

$$-\log p(X = x, PA = \boldsymbol{pa} \mid f, \sigma_x) = -\sum_{j=1}^{n}\left(E[\log p(\tilde{\epsilon}_j)] + E[\log|\det \mathbf{J}_j|]\right), \tag{11}$$

where $\tilde{\epsilon}_j$ represents $j$th dimension of estimated noise $\boldsymbol{\epsilon}$, and $\mathbf{J}_j$ is the Jacobian matrix of transformation from $(PA, X)$ to $(PA, \tilde{\epsilon}_j)$, which is calculated as

$$\det \mathbf{J}_j = \det \begin{bmatrix} \dfrac{\partial PA}{\partial PA} & \dfrac{\partial PA}{\partial X} \\ \dfrac{\partial \tilde{\epsilon}_j}{\partial PA} & \dfrac{\partial \tilde{\epsilon}_j}{\partial X} \end{bmatrix} = \det \begin{bmatrix} \mathbb{I}_m & \mathbf{0} \\ -f'(\boldsymbol{pa})^T & \boldsymbol{k}'_{(j)} \end{bmatrix} = \boldsymbol{k}'_{(j)}, \tag{12}$$

where $\mathbb{I}_m$ is an $m$-dimensional identity matrix, where $m$ is the number of variables in $PA$. $f'(\boldsymbol{pa})$ represents the derivative vector of $f$ concerning $PA$, while $\boldsymbol{k}'_{(j)}$ signifies the $j$th element in the derivative of $\boldsymbol{k}_x$ concerning $X$. More specifically, we utilize the widely-used Gaussian kernel throughout the paper, resulting in $\boldsymbol{k}'_{(j)} = -\frac{(x-x_j)}{\sigma_x^2} \cdot \exp\left[\frac{(x-x_j)^2}{2\sigma_x^2}\right]$.

It is well-known that directly maximizing the likelihood function itself may potentially lead to overfitting in structure learning (Uemura et al., 2022). To mitigate this issue, we opt to use the marginal likelihood of $p(X, PA)$ as the score function in our method. Specifically, we assume that $f$ follows a Gaussian process, denoted as $f \sim \mathcal{GP}(0, K_{PA})$, where $K_{PA}$ is the kernel matrix of $PA$ with a trainable parameter $\sigma_p$. By integrating Eq. 11 with respect to $f$, we parameterize $f$ with $\sigma_p$. Consequently, for the random variable $X$ and its independent variables $PA$ with finite observations $D$, our score function, using the marginal likelihood of joint density $p(X, PA)$ within the extended model, is as follows:

$$\begin{aligned}
S(X, PA) = &-\frac{1}{2}\mathrm{trace}\left\{K_X(K_{PA} + \sigma_\epsilon^2 I)^{-1} K_X\right\} \\
&-\frac{n}{2}\log|K_{PA} + \sigma_\epsilon^2 I| - \frac{n^2}{2}\log 2\pi + \sum_{i \neq j}^{n}\log|K'_{X(ij)}|,
\end{aligned} \tag{13}$$

where $K_X$ and $K_{PA}$ is the kernel matrix of $X$ and $PA$ on $D$, $K'_X$ is the derivative of $K_X$ respect to $x_i$, $\sigma_\epsilon$ is a hyper-parameter. See Appendix A.2 for the detailed derivations. Overall, the whole hypothetical graph $\mathcal{G}$ which has $\mathcal{Q}$ variables is scored as

$$\mathbf{S}(\mathcal{G}; D) = \sum_{q=1}^{\mathcal{Q}} S(X_q, PA_q^{\mathcal{G}}), \tag{14}$$

where $X_q$ is the variable in $\mathcal{G}$ and $PA_q^{\mathcal{G}}$ represents its independent variables in the graph.

### 3.4 DIFFERENCE WITH EXISTING METHOD

It is important to highlight that even though our score function shares a similar form with the score function in (Huang et al., 2018), with the exception of the last term, there are noteworthy distinctions in terms of causal relationship modeling. The score function that Huang et al. (2018) proposed is grounded in a regression model with pre-defined features of dependent variables and relies on the likelihood of conditional distribution. It requires the manual selection and fixing of kernel parameters beforehand. In contrast, our method models the causal relationship as a mixture of independent noise variables with learnable kernel functions, offering greater flexibility. Accordingly, we utilize the joint distribution likelihood of the involved variables as the score function to enforce constraints on the additional parameters. Consequently, with trainable kernels, our score function can automatically learn the optimal kernel for the given data within our extended model. This flexibility allows our method to adapt to more complex scenarios and accurately capture the causal relationships.

## 4 SEARCH PROCEDURE

When applying score functions to causal discovery, it is necessary to consider its properties of local consistency and score equivalence. These properties ensure that the score functions can identify the optimal causal graph or its equivalence class based on the available data.

**Local score consistency.** The proposed score function is expected to be local consistent. That is, the score of a DAG model should (1) increase when adding an edge that eliminates an independence constraint which is not presented in the generative distribution, and (2) decrease as a result of adding any spurious edge that does not eliminate such a constraint. Formally, we have the following definition of score local consistency.

**Definition 1** (Score Local Consistency (Chickering, 2002)) *Let $\mathcal{G}$ be any DAG, and let $\mathcal{G}'$ be the DAG that results from adding the edge $X_i \to X_j$ on $\mathcal{G}$. Let $D$ be the observation dataset. A score function $S(\mathcal{G}; D)$ is locally consistent if the following two properties holds as the sample size $n \to \infty$:*

1. *If $X_j \not\perp\!\!\!\perp X_i \mid PA_j^{\mathcal{G}}$, then $S(\mathcal{G}'; D) > S(\mathcal{G}; \mathcal{D})$.*

2. *If $X_j \perp\!\!\!\perp X_i \mid PA_j^{\mathcal{G}}$, then $S(\mathcal{G}; D) > S(\mathcal{G}'; D)$.*

Moreover, the following Lemma 1 shows that our score function using the marginal likelihood of involved variables is locally consistent.

**Lemma 1** *Under the condition that,*

$$\lim_{n \to \infty} \frac{1}{3!}(\boldsymbol{\sigma}_\epsilon - \widehat{\boldsymbol{\sigma}}_\epsilon)^3 \frac{\partial^3 \log p(X, PA^{\mathcal{G}} \mid \widehat{\boldsymbol{\sigma}}_\epsilon)}{\partial \boldsymbol{\sigma}_\epsilon^3} = 0, \tag{15}$$

*and with a prior that $p(\boldsymbol{\sigma}_\epsilon) = 1$ over the neighborhood of the true parameters $\widehat{\boldsymbol{\sigma}}_\epsilon$, the score function using the marginal likelihood of joint distributions is locally consistent.*

Note that $\boldsymbol{\sigma}$ comprises all the trainable parameters in the model and $\widehat{\boldsymbol{\sigma}}$ represents the true parameters with the maximum of marginal likelihood. The condition given in Eq. 15 means that the estimated parameters $\boldsymbol{\sigma}_\epsilon$ is close to the true parameters $\widehat{\boldsymbol{\sigma}}_\epsilon$ as $n \to \infty$. The proof is provided in Appendix A.3.

**Score Equivalence.** We also care about the class equivalence property of the score function to investigate whether the DAGs with the same independence constraints have the same score. Formally, score equivalence is defined as follows.

**Definition 2** (Score Equivalence (Chickering, 2002)) *Let $D$ be the observation dataset. A score function $S$ is score equivalent if for any two DAGs $\mathcal{G}$ and $\mathcal{G}'$, which are in the same Markov equivalence class, we have $S(\mathcal{G}; D) = S(\mathcal{G}'; D)$.*

Similar to the previous generalized score function, we found that using the marginal likelihood of joint distribution as score function, different DAGs in the same Markov equivalence class may have different scores, i.e. our score function is not score equivalent. The reason is that the causal relationship in our extended model is nonlinear. And with nonlinear relationships, models in opposite directions may have different scores (Hoyer et al., 2008; ZHANG, 2009).

Even though our score is not score equivalent, it is still possible to obtain the asymptotically optimal searched graph. It has been proved that using a locally consistent score function with the GES algorithm (Chickering, 2002) as the search procedure, it guarantees to find the Markov equivalence class which is consistent to the data generative distribution asymptotically, which does not require score equivalence (Huang et al., 2018). Therefore, we use the marginal likelihood of joint distribution as the score function coupled with GES as search procedure to identify the underlying causal graphs in our approach.

## 5 Experimental Results

We conducted experiments on both synthetic data and real-world benchmark datasets to evaluate the proposed score function. We compared our score function with the previous conditional likelihood-based score function methods: CV (cross-validated likelihood) and Marg (marginal likelihood) in (Huang et al., 2018). All the score-based methods were combined with GES algorithm as the search procedure for causal structure learning. Additionally, we conducted a comparison with the constraint-based search algorithm PC (Spirtes et al., 2000) and employed the kernel-based conditional independence test method (KCI) (Zhang et al., 2011) to assess independence relationships. This is denoted as PC in our comparisons.

In all experiments involving kernel-based methods, we employed the widely-used Gaussian kernel. In line with the default kernel configuration outlined in (Huang et al., 2018), we set the kernel bandwidth as twice the median distance between the original input points for both CV and Marg. We also initialized our method with this same kernel bandwidth to our methods, demonstrating that our approach can commence with heuristic initialization and learn improved kernel parameters through optimization, as compared to utilizing fixed ones. More experimental details can be found in Appendix B.2.

### 5.1 Synthetic Data

To verify the effectiveness and generality of our proposed method, we generated various types of data, including continuous, discrete, and mixed continuous-discrete data. For each variable $X_i$ in the graph, the data was generated according to:

$$X_i = g_i(f_i(PA_i) + \epsilon), \tag{16}$$

where $f_i$ was randomly chosen from the *linear*, *sin*, *cos*, *tanh* functions and their combinations, and $g_i$ was chosen from *linear*, *exponential* and *power* functions with power values ranging from 1 to 3. The noise term $\epsilon$ was randomly chosen from either a *Gaussian* or *uniform* distribution. We generated causal graphs with different graph densities ranging from $0.2$ to $0.8$. The graphs had varying numbers of variables, ranging from 6 to 10 with two sample sizes $n = 500$ and $n = 1000$. For each graph density, sample size and data type, we generated 10 realizations with different numbers of variables and data relationships.

Figure 2 presents the F1 score [1] of the recovered causal graphs using our proposed score function, along with the score-based methods CV and Marg, as well as the constraint-based PC. The x-axis shows the graph density, measured by the ratio of the number of edges to the maximum possible number of edges in the graph. The y-axis is the F1 score; higher F1 scores mean higher accuracy. In general, the F1 score decreases as the graph density increases for all methods. There are some

---

[1]F1 score is a weighted average of the precision and recall, with $F1 = 2\frac{recall \cdot precision}{recall + precision}$

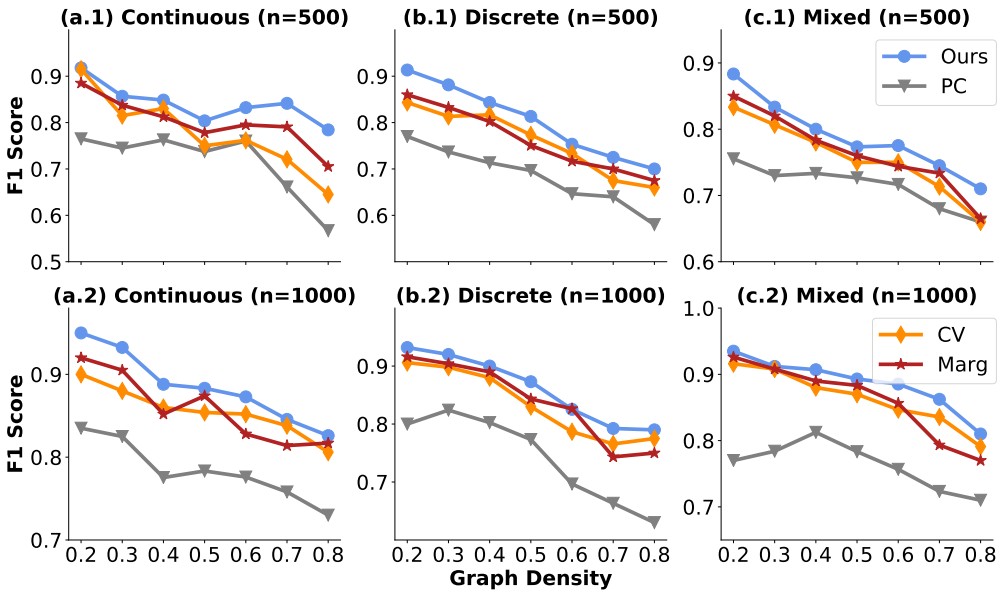

Figure 2: The F1 score of recovered causal graphs on: (a.1) Continuous data with $n = 500$; (b.1) Discrete data with $n = 500$; (c.1) Mixed data with $n = 500$; (a.2) Continuous data with $n = 1000$; (b.2) Discrete data with $n = 1000$; (c.2) Mixed data with $n = 1000$. The x-axis is the graph density and the y-axis is the F1 score; higher F1 score means higher accuracy.

fluctuations due to the randomness of the graph size as well as the generated relations. Overall, our joint distribution likelihood-based score function consistently performs the best in all settings, particularly when the graphs are dense, which demonstrates the effectiveness of our score function in capturing the underlying causal relationships. Compared with the pre-defined kernel-based CV and Marg, the improvement that our method achieved in F1 score highlights the advantages of using a trainable kernel with the marginal likelihood of joint distribution as score function in capturing the causal relationships.

We also exploited another accuracy measurement, the normalized structural hamming distance (SHD) (Tsamardinos et al., 2006), to evaluate the difference between the recovered Markov equivalence class and the true class with correct directions. Figure 3 gives the normalized SHD of the recovered Markov equivalence class. A lower SHD score indicates better performance in graph recovery. Overall, we found that our kernel selection method gives the best accuracy in all cases, which is consistent with the results measured by F1 score. The experimental results on synthetic data highlight the significance of appropriate kernel selection and the advantages of our approach. By automatically selecting the optimal kernel that best fits the data, our method can better capture causal relationships and achieve more accurate causal discovery.

## 5.2 BENCHMARK DATASETS

We also evaluated our proposed kernel selection method on two benchmark datasets commonly used in causal discovery: SACHS and CHILD. SACHS dataset consists of 11 variables, while the CHILD dataset has 20 variables. All variables in both datasets are discrete, with cardinalities ranging from 1 to 6. For each benchmark dataset, we randomly chose the data with sample size $n = 200, 500, 1000$ and 2000, and repeated 20 times in each sample size.

Figure 4 gives the F1 score and SHD score of the recovered causal skeleton on SACHS and CHILD. Overall, the F1 score increases with the sample size while the SHD score decreases for all the methods, illustrating that the accuracy increases with larger sample size in both SACHS and CHILD dataset. On the SACHS dataset, the score-based methods (CV, Marg and Ours) outperform the constraint-based PC. Among the score-based methods, our proposed method achieves the best performance across all sample sizes. On the CHILD dataset, our method consistently outperforms the

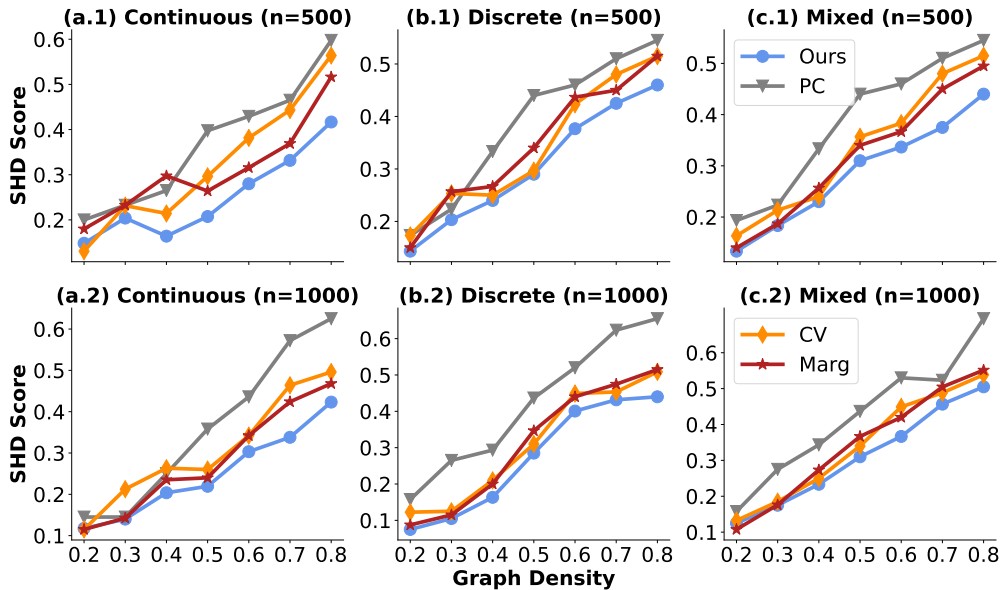

Figure 3: The normalized SHD of recovered causal graphs on synthetic data with different data types and sample sizes. The y-axis is the normalized SHD score and the lower SHD score means better accuracy.

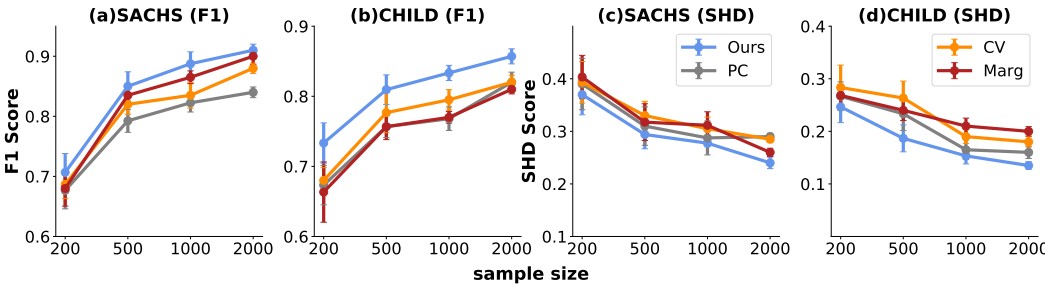

Figure 4: Performance on two popular benchmark dataset. The x-axis is the sample size. (a) and (b) are the F1 scores on SACHS and CHILD. Higher F1 score means higher accuracy. (c) and (d) are the SHD scores; the lower SHD score means better accuracy.

other methods by a significant margin. The experimental results on the benchmark datasets highlight the advantages of our method in real-world scenarios. It also indicates that manually-selected kernels may not be suitable for many real-world data and the automatic selection of the optimal kernel in our method enables better handling of the wide range of causal relationships in the real world.

## 6 CONCLUSION

In this paper, we presented a novel kernel selection method in the generalized score function for causal discovery, which allows for the automatic selection of the optimal kernel that best fits the data. We extend the regression model in RKHS and model the causal relation between variables as a mixture of independent noise variables. Base on this model, we treat the kernel as trainable parameters that can be optimized along with other parameters in the model. Accordingly, we utilize the marginal likelihood of joint distribution as the score function to automatically select the optimal kernel specific to the given data. Experimental results on both synthetic data and benchmark datasets demonstrated the effectiveness of our method in automatically selecting superior kernels compared to manually selected ones and its contributions on precise relationship characterization and accurate causal discovery.

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

# Appendix

In this section, we present all the proofs and more experimental details and results. The content of the Appendix is as follows:

- **Proofs**
  
- **More experimental details and results**
  

## A  PROOFS

### A.1  THE RELATIONSHIP BETWEEN MUTUAL INFORMATION AND LIKELIHOOD

According to Eq.7, the random variables $X$ and $PA$ hold the following relation

$$\boldsymbol{k}_x = f(PA) + \boldsymbol{\epsilon}, \tag{17}$$

where $\boldsymbol{k}_x$ represents the feature of $X$ in the RKHS, $f$ denotes a nonlinear function, and $\boldsymbol{\epsilon}$ represents the noise term, which is independent of $PA$. Considering a dataset of $n$ observations, denoted as $D = \{(x_1, \boldsymbol{pa}_1), (x_2, \boldsymbol{pa}_2), \cdots, (x_n, \boldsymbol{pa}_n)\}$, the feature vector $\boldsymbol{k}_x$ becomes an $n$-dimensional vector, given by $\boldsymbol{k}_x = (k_1, \cdots, k_n)$. Additionally, the noise term is defined as $\boldsymbol{\epsilon} = (\epsilon_1, \cdots, \epsilon_n)$, with each element $\epsilon_i$ being independent of the others.

Suppose that the noise term $\epsilon_i \sim \mathcal{N}(0, \sigma_\epsilon^2)$. According to Eq.5, we denote $k_i = g(X)$. Based on the change-of-variables theorem, the joint density $p(X, PA)$ can be estimated by

$$
\begin{aligned}
p(X, PA; f, g) &= p(X, \tilde{\epsilon}_i; f, g) \cdot |\det \mathbf{J}_i| \\
&= \frac{1}{\sqrt{2\pi\sigma_\epsilon^2}} \exp\left(-\frac{(g(X) - f(PA))^2}{2\sigma_\epsilon^2}\right) \cdot |\det \mathbf{J}_i|,
\end{aligned} \tag{18}
$$

where $\tilde{\epsilon}_i = g(X) - f(PA)$ is estimated noise and $\mathbf{J}_i$ is the Jacobian matrix of transformation from $(PA, X)$ to $(PA, \tilde{\epsilon}_i)$, i.e. $\mathbf{J}_i = [\partial(PA, \tilde{\epsilon}_i)/\partial(PA, X)]$. Therefore, we can obtain the log-likelihood of $p(X, PA)$ on $D$:

$$
\begin{aligned}
\log L(f, g\,; D) &= \log \prod_{j=1}^{n} p(x_j, \boldsymbol{pa}_j; f, g) \\
&= \sum_{j=1}^{n} \left\{ \log \frac{1}{\sqrt{2\pi\sigma_\epsilon^2}} \exp\left(-\frac{(g(x_j) - f(\boldsymbol{pa}_j))^2}{2\sigma_\epsilon^2}\right) + \log|\det \mathbf{J}_i|_{(j)} \right\} \\
&= n \left\{ E[\log p(\tilde{\epsilon}_i)] + E[\log|\det \mathbf{J}_i|] \right\},
\end{aligned} \tag{19}
$$

Therefore, the log likelihood under the extended model in Eq.7 is

$$
\begin{aligned}
\log L(\sigma_x, f\,; D) &= \sum_{i=1}^{n} n \left\{ E[\log p(\tilde{\epsilon}_i)] + E[\log|\det \mathbf{J}_i|] \right\} \\
&= n \left\{ E[\log p(\tilde{\boldsymbol{\epsilon}})] + E[\log|\det \mathbf{J}|] \right\}
\end{aligned} \tag{20}
$$

where we represent $|\det \mathbf{J}| = \sum_{i=1}^{n} |\det \mathbf{J}_i|$ and $\sigma_x$ is the parameter of the kernel function. As can be seen, minimizing the mutual information between $X$ and $PA$ in Eq.10 is equivalent to maximizing the likelihood of $p(PA, X)$ under our extended model in Eq.20.

## A.2 Derivation of marginal likelihood of joint distribution in RKHS

Suppose that the estimated noise is Gaussian, denoted as $\epsilon \sim \mathcal{N}(0, \sigma_\epsilon^2 I)$ and $f$ is a Gaussian process with $f \sim \mathcal{N}(0, K_{PA})$. According to Eq.18, for one particular observation $(x, \boldsymbol{pa}) \in (\mathcal{X}, \mathcal{P})$, the log marginal likelihood of $p(x, \boldsymbol{pa})$ is

$$
\begin{aligned}
\log p(X = x, PA = \boldsymbol{pa} \mid \sigma_x, \sigma_p) &= \log \int p(x, \boldsymbol{pa} \mid f, \sigma_x) \cdot p(f \mid \sigma_p) \, \mathrm{d}f \\
&= \log \int p(x, \boldsymbol{pa} \mid f, \sigma_x) \cdot p(f \mid \sigma_p) \, \mathrm{d}f + \sum_{j=1}^n \log \int |\det \mathbf{J}_j| \cdot p(f \mid \sigma_p) \, \mathrm{d}f \\
&= \log \int p(x, \boldsymbol{pa} \mid f, \sigma_x) \cdot p(f \mid \sigma_p) \, \mathrm{d}f + \sum_{j=1}^n \log |\det \mathbf{J}_j| \\
&= \log \int \mathcal{N}(0, \sigma_\epsilon^2 I) \cdot \mathcal{N}(0, K_{PA}) \, \mathrm{d}f + \sum_{j=1}^n \log |\det \mathbf{J}_j| \\
&= \log \mathcal{N}(0, K_{PA} + \sigma_\epsilon^2 I) + \sum_{j=1}^n \log |\boldsymbol{k}'_{(j)}|
\end{aligned}
$$

where $\sigma_p$ is the parameter in $K_{PA}$. Let us further denote $K_\theta = (K_{PA} + \sigma_\epsilon^2 I)$. With $n$ observations $D = \{(x_1, \boldsymbol{pa}_1), (x_2, \boldsymbol{pa}_2), \cdots, (x_n, \boldsymbol{pa}_n)\}$, the log likelihood of joint density $(X, PA)$ on $D$ is represented as

$$
\begin{aligned}
\log l(\sigma_x, \sigma_p) &= \prod_{i=1}^n \left\{ \log \mathcal{N}(0, K_\theta) + \sum_{j=1}^n \log |\boldsymbol{k}'_{(j)}| \right\} \\
&= -\frac{1}{2} \sum_{i=1}^n \boldsymbol{k}_{x(i)}^T K_\theta^{-1} \boldsymbol{k}_{x(i)} - \frac{n}{2} \log \det |K_\theta| - \frac{n^2}{2} \log 2\pi + \sum_{i=1}^n \sum_{j=1}^n \log |\boldsymbol{k}'_{x(ij)}| \\
&= -\frac{1}{2} \sum_{i=1}^n (K_X K_\theta^{-1} K_X)_{ii} - \frac{n}{2} \log \det |K_\theta| - \frac{n^2}{2} \log 2\pi + \sum_{i \neq j} \log |K'_{X(ij)}| \\
&= -\frac{1}{2} \mathrm{trace}(K_X K_\theta^{-1} K_X) - \frac{n}{2} \log \det |K_\theta| - \frac{n^2}{2} \log 2\pi + \sum_{i \neq j} \log |K'_{X(ij)}|
\end{aligned}
$$

where $i$ represents the $j$th sample in $D$ and $j$ is the $j$th element in the feature vector $\boldsymbol{k}_x$ Therefore, $\boldsymbol{k}'_{x(ij)} = K'_{X(ij)}$ where $K_X$ and $K_{PA}$ is the kernel matrix of $X$ and $PA$ with the leranable parameters $\sigma_x$ and $\sigma_p$ respectively.

A.3   PROOF OF LEMMA 1

Our score function using marginal likelihood of joint density $(X_i, PA_i^{\mathcal{G}})$ is

$$S(X_i, PA_i^{\mathcal{G}}) = \log p(X_i, PA_i^{\mathcal{G}}|\boldsymbol{\sigma}_\epsilon), \qquad (21)$$

And $\boldsymbol{\sigma}_\epsilon$ represents the trainble parameters in the model. For example, when we choose Gaussian kernel, there are two parameters in the model: the bandwidth of the kernels applied on the dependent variable and its parents, denoted as $\boldsymbol{\sigma}_\epsilon = [\sigma_X, \sigma_{PA}]$. We define $\widehat{\boldsymbol{\sigma}}_\epsilon = [\widehat{\boldsymbol{\sigma}}_X, \widehat{\boldsymbol{\sigma}}_{PA}]$ to be the true parameters with the maximum of $p(X, PA_i^{\mathcal{G}})$. By employing the Laplace method (Maxwell Chickering & Heckerman, 1997), we can derive that

$$\log p(X_i, PA_i^{\mathcal{G}}|\boldsymbol{\sigma}_\epsilon) \approx \log p(X_i, PA_i^{\mathcal{G}} \mid \widehat{\boldsymbol{\sigma}}_\epsilon) + \log p(\widehat{\boldsymbol{\sigma}}_\epsilon) + \frac{d}{2}\log(2\pi) - \frac{1}{2}\log|A|, \qquad (22)$$

where $d$ is the number of trainable parameters in the model, and $A$ is the negative Hessian of $\log p(X_i, PA_i^{\mathcal{G}} \mid \boldsymbol{\sigma}_\epsilon)$ evaluated at $\widehat{\boldsymbol{\sigma}}_\epsilon$. The first term in Eq.22, $\log p(X_i, PA_i^{\mathcal{G}} \mid \widehat{\boldsymbol{\sigma}}_\epsilon)$ increases linearly with the sample size $n$. And $\log|A|$ increases as $d\log n$. The remaining two terms $\log p(\widehat{\boldsymbol{\sigma}}_\epsilon)$ and $\frac{d}{2}\log(2\pi)$ are both consistent with $n$ increasing. That is, for a large sample size $n$, we obtain

$$\log p(X_i, PA_i^{\mathcal{G}} \mid \sigma_\epsilon) \approx \log p(X_i, PA_i^{\mathcal{G}} \mid \widehat{\boldsymbol{\sigma}}_\epsilon) - \frac{d}{2}\log n. \qquad (23)$$

This approximation is equivalent to the Bayesian Information Criterion (BIC) (Schwarz, 1978). Since BIC is consistent (Haughton, 1988), the marginal likelihood of joint distribution $p(X_i, PA_i^{\mathcal{G}})$ as score function is also consistent.

Consequently, we have demonstrated the local consistency of our proposed score under the conditions outlined in Lemma 1. Consider a variable $X$ and all its potential parent variable combinations, denoted as $[PA_1, PA_2, \cdots, PA_n]$. Each combination $PA_i$ with $X$ corresponds to its respective maximum marginal likelihood of $p(X, PA_i \mid \widehat{\boldsymbol{\sigma}}_\epsilon)$, denoted as $S_i$. The local consistency property guarantees that among all these combinations, the score for the correct $PA_{true}$ variable for $X$, denoted as $S_{true}$, is the highest. Therefore, our proposed score function is effective in identifying the correct $PA$ variables of $X$ from among all the potential parents.

# B   MORE EXPERIMENTAL DETAILS AND RESULTS

## B.1   VISUALIZATION OF MORE DIMENSIONS OF THE CASES IN MOTIVATION

In the Motivation Section, we conducted an experiment involving two cases where conditional likelihood-based score function failed to learn the correct structure. We generated a total of $n = 500$ points as the observation data for each case. Hence, each observation $x$ is associated with a feature $\boldsymbol{k}_x$ in RKHS with $500$ dimensions. We utilized the Adam optimizer Kingma & Ba (2014) for the training of parameters in the projection $f$ in Equation 1 during each search procedure, and the model was trained for a total of 500 iterations.

Figure 5 and Figure 6 depict the visualization of additional dimensions. The figures show that the remaining dimensions of the feature exhibit a similar pattern to the first dimension, as depicted in Figure 1 in the paper.

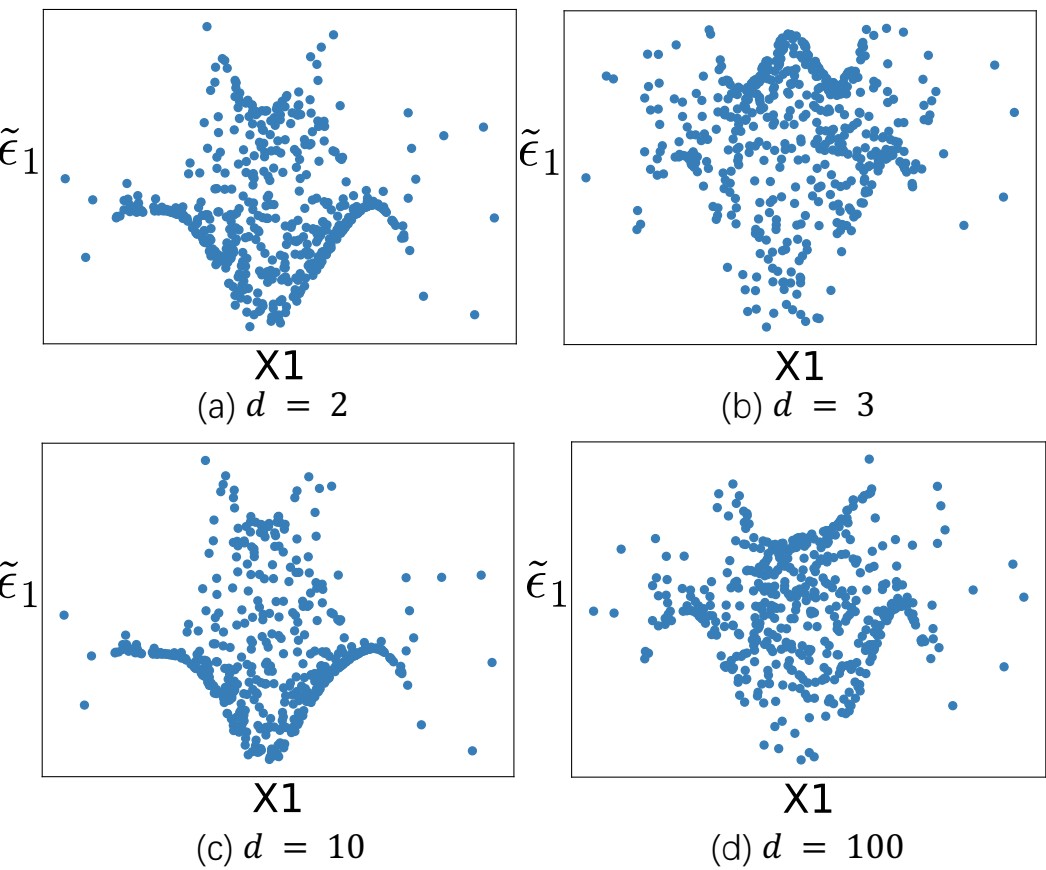

(a) $d = 2$        (b) $d = 3$

(c) $d = 10$        (d) $d = 100$

Figure 5: The visualization of the rest dimensions in cases 1. $d$ represents the $d$th dimension of the feature $\boldsymbol{k}_x$.

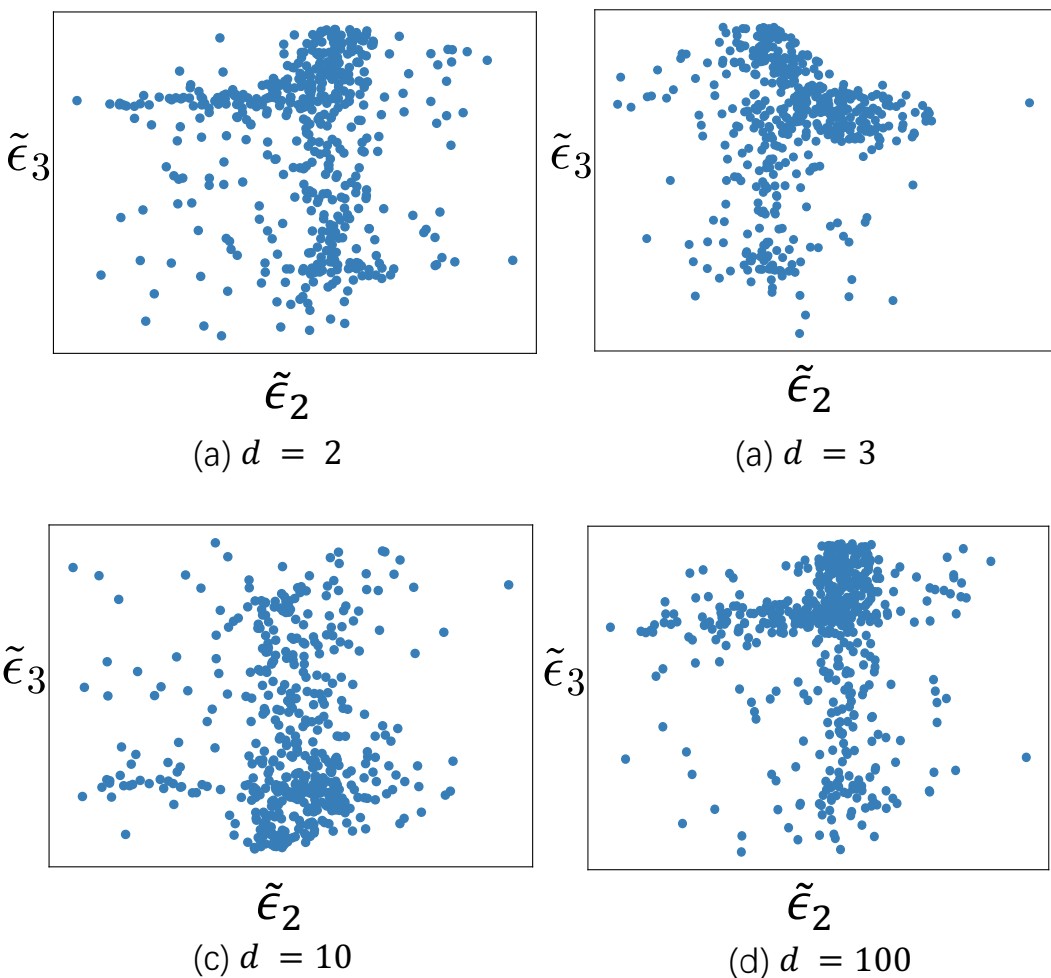

Figure 6: The visualization of the rest dimensions in cases 2. $d$ represents the $d$th dimension of the feature $\boldsymbol{k}_x$.

## B.2 EXPERIMENTAL DETAILS

We present the implementation details for the learning procedure of our methods. The hyper-parameter for noise scale $\sigma_\epsilon$ is uniformly set to $0.01$ for Marg, CV, and our method. During each search step, our model undergoes training for 1000 iterations. We employed the Adam optimizer (Kingma & Ba, 2014) with an initial learning rate of 0.1 to optimize the parameters in our method.

To mitigate the influence of noise patterns arising from limited data, we introduce a threshold $\tau$ for local changes in GES. Specifically, our method executes the corresponding operation in the graph only when the score change before and after applying the Insert or Delete operator exceeds the threshold $\tau$. For synthetic continuous data, $\tau$ is set to 0.002, while for synthetic discrete and mixed data, it is set to 0.01. As for the real benchmark datasets, we set $\tau$ to $0.8$ for SACHS and $0.5$ for CHILD. The code of all the methods is highly based on the *causal-learn* toolbox, which provides implementations of the compared methods.

