# OpenReview forum: "Optimal Kernel Choice for Score Function-based Causal Discovery"
_ICLR.cc/2024/Conference — Submitted to ICLR 2024_

### Official Review · Reviewer_VHXH · 2023-10-13

**Soundness:** 2 fair
**Presentation:** 1 poor
**Contribution:** 1 poor
**Rating:** 5
**Confidence:** 4

**Summary:**

Authors propose a Gaussian-process-based method for Bayesian network structure learning. They develop a score criterion based on marginal likelihood to optimize the kernel hyper parameter. However, the significance of this score criterion is unclear because the proposed idea is similar to that of UAI2000 paper [1], which employs Gaussian processes for structure learning and introduces several score criteria for optimizing the kernel hyper parameter. Since the authors do not even cite this paper, the novelty and the significance are unclear to me.

[1] Nir Friedman, Iftach Nachman. “Gaussian Process Networks.” Proceedings of the Sixteenth Conference on Uncertainty in Artificial Intelligence (UAI2000).

**Strengths:**

- The proposed score criterion is founded on the recent idea of generalized score criterion [Huang+; KDD2018].

**Weaknesses:**

(A) The difference from the existing Gaussian-process-based framework is unclear.

- As described above, the idea of optimizing kernel hyper parameters in the context of Bayesian network structure learning has already been developed more than 20 years ago.
- What are the key differences from UAI2000 paper [1]?
- I am not sure, but isn’t the proposed score function in Eq. (13) very similar to that of [1]?

(B) Notations are unclear and seem inconsistent, which makes the paper hard to follow.

- In particular, the notations around Eq. (1) are unclear. Noise $\varepsilon$ is introduced without definition. Is it a function in RKHS, i.e., $\varepsilon \in \mathcal{H}_X$? If so, state it clearly. Definition of $k_x$ is unclear: $x$ is a realization of random variable $X$? What is the relationship with kernel feature mapping $\phi_X$? $\phi_X(x) = k_x$? Is it defined in the same way as Section 3.1?

- (In causal inference, usually, the parent set of random variable is denoted by e.g., $PA_X$, not $PA$, to clearly state the child variable.)

- The connection between Eqs. (3) and (4) is unclear. I believe that authors express $\phi_i(x)$ as the i-th element of (possibly) infinite-dimensional feature vector $\phi_X(x)$, where $\phi_X \in \mathcal{H}_X$. However, they say that $g(x) = \phi_i(x)$ also belongs to the same RKHS $\mathcal{H}_X$. What does this mean? Please clearly define what indices $i$ and $j$ mean.

- In Eq. (6), there is no definition or description about parameter $\theta$. What kind of parameters do the authors consider?


Typo: See the beginning of Section 2 carefully.

**Questions:**

- As described above, the idea of optimizing kernel hyper parameters in the context of Bayesian network structure learning has already been developed more than 20 years ago.
- What are the key differences from UAI2000 paper [1]?
- I am not sure, but isn’t the proposed score function in Eq. (13) very similar to that of [1]?

---

> ### Author Response · Authors · 2023-11-20
>
> Dear Reviewer VHXH,
>
> Thank you very much for your efforts in reviewing this paper. We identify there exist some misunderstandings. And we have reexamined the notation in the paper fixing the typos. Here are some clarification of your concerns.
>
> **Q1. The difference from the existing Gaussian-process-based framework is unclear.**
>
> **A1.** I believe there might be some misunderstanding here. The contribution of our method does not lie in using Gaussian processes to optimize kernel hyperparameters.
> Our contribution builds upon Huang et al.'s method.
> Huang et al. utilize the characterization of general independence relationships with conditional covariance operators and propose using the regression model for causal discovery with the regression form $\phi(X; \sigma_x) = f(PA; \theta) + \epsilon$.
> In Huang et al.'s model, **the parameter $\sigma_x$ in $\phi(\cdot)$ is pre-defined**.
> It assumes that the pre-defined feature $\phi(X)$ is generated from a non-linear projection $f(PA_X)$ with an independent noise $\epsilon$ added, which is hard to satisfy with such an manually selected $\phi(\cdot)$ and may not fully capture the characteristics of the data in many cases.
> While in our method, we extended the regression model above with trainable parameters of $\phi(\cdot)$ rather than the fixed one.
>
> By introducing more learnable parameters, our model becomes more flexible and can more accurately uncover relationships within the data. Additionally, due to the introduction of extra parameters, we propose using the joint distribution's marginal likelihood as the score for parameter learning and causal discovery, which is different from Huang's conditional likelihood based ones.
> To the best of our knowledge, there is currently no other approaches using the same model and score function for causal discovery.
>
> And our proposed score function in Eq.13 is not the same with that of [1].
> **Our score has an extra term (last term in Eq.13), which reflects the difference of relationship modeling mention above.**
> And this last term arises from transforming the joint distribution of dependent and independent variables into the joint distribution of the independent variables and the estimated mixed noise.
> It results from simplifying the Jacobian matrix obtained from using the change-of-variables theorem.
>
> **Q2. Relationship between $k_x$ and $\phi(x)$**
>
> **A2.** When we introduce an RKHS space $H_\mathcal{X}$ on $\mathcal{X}$, it comes with the definition of a continuous feature mapping $\phi_\mathcal{X}$ and a measurable positive-definite kernel function $k_\mathcal{X} : \mathcal{X} \times \mathcal{X} \rightarrow \mathbb{R}$.
> Since $\phi_X$ is in an infinite-dimensional space without a proper probability measure.
> For practical purposes, given an observation $x$ of $X$, we project $\phi_\mathcal{X}(x)$ into its empirical feature map, which is $\mathbf{k}_x$.
>
> **Q3. The connection between Eq.3 and Eq.4 is unclear.**
>
> **A3.** We are sorry for your confusion.
> The Eq.3 is derived from Theorem 7 in [1], which is the theoretical foundation that allows us to transform the independence relations test into a regression model selection problem.
> Eq.4 is the corresponding regression model, where a continuous feature mapping $\phi(X)$ is applied to the dependent variable $X$.
> Due to the properties of characteristic kernels, $\phi(X)$ can be decomposed into an infinite linear combination of basis functions $[\phi_1, \cdots, \phi_i, \cdots]$, and each $\phi_i \in {\mathcal{H_X}}$ is orthogonal to each other.
> We associate each $\phi_i$ with the function $g$ in Eq.3, leading to the conclusion that Eq.5 holds for any given $\phi \in {\mathcal{H_X}}$.
> Thus, based on Eq.5, we can further transform the conditional independence test into a parameter selection problem for the regression model defined in Eq.4.
>
> **Q4. In Eq.6, there is no definition or description about parameter. What kind of parameters do the authors consider?**
>
> **A4.** In the later part, we introduced Gaussian processes to mitigate model overfitting. By parameterizing $f$, we aim to reduce the number of parameters in $f$, representing the parameters $\theta$ as optimizable parameters in the Gaussian process.
>
> ### Reference
>
> [1] K. Fukumizu, F. R. Bach, and M. I. Jordan. 2004. Dimensionality reduction for supervised learning with reproducing kernel Hilbert spaces. Journal of Machine Learning Research 5 (2004), 73–79.

---

> ### Comment · Reviewer_VHXH · 2023-11-21
> **Not all my questions are answered**
>
> Unfortunately, I cannot clearly understand what the authors mean, mainly due to the grammatical errors in their response, though I will not enumerate them.
>
> I posed two questions about notations,  $\varepsilon \in H_X$? and $\phi(x) = k_x$?, but they seem not be fully answered (I understand the bold $k_x$ is a concatenated feature vector of observations; thank you for the response.). Regarding the latter, in usual setup in kernel methods, we often express feature mapping $\phi$ like as $\phi(x) = k_{X}(x, \cdot)$. However, from the authors' response, I am not sure whether the notations are introduced in the same way.
>
> As pointed out by other Reviewer, if the related work is not sufficiently mentioned, the readers cannot understand the technical significance of the proposed method. I understand that developping GP-based formulation for Bayesian network structure learning like UAI2000 paper is not the proposal of this work. However, since both address similar problem setup and use similar GP-based formulations, the authors should clearly explain the difference. Although the authors mention the formulation difference in their response (i.e., the last term in Eq. 13), I cannot clearly understand the meaning of *reflects the difference of relationship modeling mention above* (mention -> mentioned?; What does *relationship modeling* mean?). If they claim that their model formulation is more flexible than UAI2000 paper, they should perform empirical performance comparison.
>
> I am not sure, but I remember that UAI2000 work can also adapt the empirical Bayes framework for hyperaparameter tuning. If so, the contribution of the proposed method is to make it possible to perform gradient-based optimization for hyperparameter tuning (to my understanding). However, without any empirical comparison, we cannot see the technical significance of this contribution.
>
> For this reason, I feel that the paper requires a major revision and is not ready for the publication at such top conferences as ICLR.

---

> > ### Author Response · Authors · 2023-11-22
> >
> > The noise $\epsilon \in \mathcal{H_X}$ and we indeed intend to express $\phi(x) = k(x, \cdot)$. And we will reorganize the paper in the future version to make the description of the RKHS background more complete.
> > The 'relationship model' refers to the assumption of a generative relationship between the variable $X$ in the Given data and its corresponding dependent variable $PA_X$, i.e., conditional densities $P(X|PA_X)$.
> > In [1], the model assumes $X = f(PA_X) + \epsilon$, while in our method, we assume $\phi(x) = f(x) + \epsilon$. By introducing an additional kernel function on $X$, our method can capture more complex relationships.
> >
> > Here are some experimental results compared to UAI2000. We generated a continuous data graph and the relationships between variables are in accordance with Eq.16 in the paper.
> > For each graph density, we generated 10 cases.
> > Due to the introduction of additional kernel function applied on the dependent variable, we need to address the issue that the conditional likelihood cannot effectively supervise the parameters of $\phi(x)$.
> > Therefore, we propose the corresponding mutual information-based score function.
> >
> > | Graph density  | 0.2          | 0.4          | 0.5          | 0.6          | 0.8          |
> > |---------|--------------|--------------|--------------|--------------|--------------|
> > | Ours    | 0.950 (+0.10)| 0.776 (+0.08)| 0.887 (+0.05)| 0.768 (+0.05)| 0.865 (+0.03)|
> > | UAI2000 | 0.739 (+0.13)| 0.610 (+0.09)| 0.746 (+0.03)| 0.638 (+0.04)| 0.719 (+0.05)|
> >
> > We would like to express our gratitude once again for your review of our paper. We will make revisions based on the suggestion you provided.

---

### Official Review · Reviewer_SijP · 2023-10-25

**Soundness:** 2 fair
**Presentation:** 2 fair
**Contribution:** 2 fair
**Rating:** 5
**Confidence:** 3

**Summary:**

This paper presents an extension to score-based causal discovery with RKHS regression. The authors propose utilizing the marginal likelihood in the RKHS of dependent and independent variables as a local score function. Additionally, they suggest optimizing kernel hyperparameters by maximizing the marginal likelihood, aiming to eliminate the necessity of manual parameter selection.
The authors provide proof for the local consistency of the proposed scores, although I am a bit uncertain whether it is limited to Gaussian noise and a Gaussian kernel. In terms of empirical evaluation, the paper offers a comparison between the proposed method and baselines using manually selected kernels and conditional likelihood score functions. This comparison is conducted on both synthetic and real datasets, with the results indicating that the proposed method shows consistent improvements over the baselines.

**Strengths:**

The strength of this paper lies in its introduction of a methodology for optimizing kernel parameters, which effectively sidesteps the necessity of manual selection. The intuition of this approach is presented in an accessible manner. While the motivation behind the proposed score function could benefit from further elucidation, it is, nonetheless, founded on principled considerations.
The author fortifies the theoretical foundation of the paper with a proof of local consistency for the proposed score. Complementing the theoretical contributions, the paper includes initial empirical experiments that lend support to the claims made, although these might benefit from further expansion and depth.

**Weaknesses:**

The primary concerns of this approach pertain to the method's comparative effectiveness and certain areas requiring additional clarification.

Initially, the author posits the use of joint marginal likelihood as a score function, highlighting its ability to mitigate overfitting issues commonly associated with conditional likelihood. While this is a crucial point, it could be significantly strengthened by including experimental comparisons with a baseline where kernel hyperparameters are optimized using conditional likelihood. Additionally, a more in-depth exploration of why joint marginal likelihood successfully avoids overfitting—potentially due to the marginalization of
$f$ would contribute to a clearer understanding of the method’s advantages. A term-by-term comparison between the equations of these two score functions could provide valuable insights into the sources of their respective strengths and weaknesses.
The use of Gaussian process for $f$ is noted as advantageous for preventing overfitting and negating the need for explicit training of f. However, this brings with it a computational complexity of $O(n^3)$?

In terms of empirical validation, the paper focuses on demonstrating the learning of simple bandwidths. To underscore the method’s versatility and general applicability, consideration of more complex kernels, such as deep kernels, would be beneficial.

On the matter of clarity, questions arise regarding the method’s performance under non-Gaussian noise conditions and whether Equation 13 can be derived in such scenarios. Furthermore, for Lemma 1, it would benefit from a clarification on whether a Gaussian kernel is assumed and how the condition in Equation 15 is utilized.

**Questions:**

Apart from the weakness, I have the following questions:

1. Is Equation 11 the same as the conditional likelihood?

2. You mentioned "maximizing the likelihood function itself may potentially lead to overfitting in structure learning", what do you mean by this? Could you elaborate about it?

3. What do you mean by "mixture of independent noise variables"?

4. Why joint distribution likelihood enforces constraints on the additional parameters? Since this is the key contributions, the author should consider explaining it more clearly.

---

> ### Author Response · Authors · 2023-11-20
>
> Dear Reviewer SijP,
>
> Thank you very much for your efforts in reviewing this paper. We identify there exist some misunderstandings. And here are some clarification of your concerns.
>
> **Q1. Why joint marginal likelihood successfully avoids overfitting?**
>
> **A1.** What we want to highlight here is that **using the likelihood of the joint distribution instead of conditional distribution is not to mitigate overfitting**. Given that our model introduces additional learnable parameters (kernel function parameters imposed on the dependent variable), we choose to employ the likelihood of the joint distribution **to constrain these parameters and learn a non-trivial relationship**.
> As mentioned in the last paragraph of Section 3.2, if we were to use conditional likelihood for learning these extra parameters, the model would directly fail to capture real relationships but project all samples into a very small range of the feature space with excessively high scores, which is uninformative for causal discovery purposes.
> This is the reason why we must use the joint distribution—it fundamentally stems from the differences in the assumed models.
> Building on this, to alleviate model overfitting, we utilized Gaussian processes to parameterize the projection function $f$, aiming to enhance the effectiveness of causal discovery.
> The corresponding score also takes the form of joint marginal likelihood.
>
> **Q2. Using more complex kernels, such as deep kernels.**
>
> **A2.** Your valuable insights are appreciated. Due to the fact that our approach is based on the characterization of conditional independence with conditional covariance operators[1], it requires the use of the characteristic kernel. The use of more complex deep kernels, etc., will need further exploration.
>
> **Q3. Is Equation 11 the same as the conditional likelihood?**
>
> **A3.** It is not the same with the conditional likelihood. If it is a conditional distribution, in Equation (11), only the first term will be present, and there won't be the term $E[\log | det J_i|]$. That is, $p(x|pa, f, \sigma_x) = p(f(pa) + \widetilde{{\pmb \epsilon}}| pa, f, \sigma_x) = p(\widetilde{\epsilon})$.
>
> **Q4. You mentioned "maximizing the likelihood function itself may potentially lead to overfitting in structure learning", what do you mean by this? Could you elaborate about it?**
>
> **A4.** When using models like MLPs to represent $f$ in the model and directly employing MLE or minimizing mutual information to learn the model parameters, overfitting is prone to occur.
> The main drawback is that this model can fit perfectly to any data by learning constant functions.
> Consequently, the estimated noise $\tilde{\epsilon}$ will be constant and thus always independent from $X$ with a high score, leading to incorrect connections.
> Therefore, here we employ Gaussian Processes to parameterize $f$, with the expectation of reducing model overfitting.
> There are also some other methods to avoid overfitting, such as rank-based [2] or auto-encoding-based [3].
>
> **Q5. What do you mean by "mixture of independent noise variables"?**
>
> **A5.** We formulate the generative relationship among presumed variables as $\phi(x) = f(pa) + \epsilon$, where $\epsilon$ represents the independent noise variable vector, and each dimension is independent of the others. Subsequently, our objective involves learning the model parameters and conducting a causal graph search by fitting the likelihood of this set of independent noise variables. That is "mixture of independent noise variables".
>
> **Q6. Why joint distribution likelihood enforces constraints on the additional parameters?**
>
> **A6.** As we answered in A3, according to Eq.11, the conditional probability distribution cannot constrain the kernel function parameters imposed on $X$.
> In contrast, the joint distribution likelihood, with the inclusion of the term
> $E[\log|\mathrm{det} J_i|]$, provides constraints on the kernel function parameters imposed on $X$.
> This term plays a role in constraining the kernel function parameters by influencing the Jacobian determinant's expectation, providing valuable constraints during the parameter learning process.
>
> ### Reference
>
> [1] K. Fukumizu, F. R. Bach, and M. I. Jordan. 2004. Dimensionality reduction for supervised learning with reproducing kernel Hilbert spaces. Journal of Machine Learning Research 5 (2004), 73–79.
>
> [2] Keropyan G, Strieder D, Drton M. Rank-Based Causal Discovery for Post-Nonlinear Models International Conference on Artificial Intelligence and Statistics, 2023: 7849-7870.
>
> [3] Uemura K, Shimizu S. Estimation of post-nonlinear causal models using autoencoding structure. International Conference on Acoustics, Speech and Signal Processing, 2020: 3312-3316.

---

> > ### Comment · Reviewer_SijP · 2023-11-20
> >
> > For the Jacobian matrix, it is $\partial (PA, \tilde{\epsilon})/\partial (PA,X)$, which is equivalent to $[[I, -df/dPA],[0,I]]$? So the Jacobian is identity?

---

> > > ### Author Response · Authors · 2023-11-21
> > >
> > > Thank you for describing your concern. In the paper, we assume the relation model as $\phi(X) = f(x) + \epsilon$, where we apply a continuous function to the dependent variable $X$. In practice, we map the infinite-dimensional $\phi(X)$ to an empirical space, obtaining an $n$-dimensional vector $k_x$, where $n$ is the number of observations.
> > > Thus, in the model $\vec{k_x} = f(x) + \vec{\epsilon}$,  we assume that $\vec{\epsilon}$ follows a distribution
> > > $N(0, {\sigma}^2_\epsilon \vec{I})$ and obtain its negative log-likelihood, i.e., Eq.11 in the paper, and each dimension $\widetilde \epsilon_j$ follows Eq. 12.
> > > Therefore, in the Jacobian matrix, we have ${\partial {\widetilde \epsilon_j }} / {\partial X}= {\partial k_x(j)} / {\partial X} = k'_{(j)}$,
> > > which involves the kernel function parameters applied to $X$ and is not equal to 1. Please feel free to discuss with us if you have any concerns.

---

> > > > ### Comment · Reviewer_SijP · 2023-11-22
> > > >
> > > > Thanks for the authors' response and clarification regarding the Jacobian matrix computation.
> > > >
> > > > However, as pointed by other reviews, the contribution and effectiveness of the proposed approach is not demonstrated clearly through the experiment section. For example, only the bandwidth is learned through the proposed approach but its effectiveness can be demonstrated more clearly through the deep kernel. Although typical deep kernel is not guaranteed to be characteristic, some approach can be taken to empirically ensure that, like [1].
> > > >
> > > > Therefore, I will remain my current rating.
> > > >
> > > > [1] Li, Chun-Liang, et al. "Mmd gan: Towards deeper understanding of moment matching network." Advances in neural information processing systems 30 (2017).

---

### Official Review · Reviewer_oMj1 · 2023-10-30

**Soundness:** 2 fair
**Presentation:** 1 poor
**Contribution:** 2 fair
**Rating:** 3
**Confidence:** 4

**Summary:**

The authors propose a score-based method for causal structure learning. The method is based on a previously introduced method by Huang et al. (2018) which exploits regression in RKHS to capture statistical dependencies between variables and extends it via introducing trainable kernel hyperparameters. In an experimental validation, the authors demonstrate that their approach outperforms the method of Huang and the common PC algorithm.

**Strengths:**

- The proposed method is convincingly motivated from an information-theoretic point of view where the goal is to minimize the mutual information (MI) between parents of a variable and a noise term.
- The proposed method outperforms other score-based causal discovery methods on synthetic and two real-world data sets.
- The paper is easy to follow.

**Weaknesses:**

- The main contribution of the paper seems to be the introduction of a trainable kernel hyper-parameter (instead of previous work that estimates it heuristically from data) and an extended objective that minimizes the MI between parents of a node and noise variables. This seems too incremental as a conference contribution and in my (the reviewer's) opinion is of insufficient originality.
- The paper is extremely close to Huang et al. (2018) [1], both in writing and content as well as methodologically.
- The experimental section is very thin. Since the method by Huang et al. (2018) has been published, a multitude of other score-based methods have been introduced, against which the author's method should be compared. To name a few:
[2-4] (and references therein) all of which have been evaluated on the Sachs data.
- The paper misses several key references in the field, e.g., [2-4], but several others as well, e.g., [5-7].

[1] https://www.kdd.org/kdd2018/accepted-papers/view/generalized-score-functions-for-causal-discovery
[2] https://arxiv.org/abs/2105.11839
[3] https://arxiv.org/abs/1909.13189
[4] https://arxiv.org/abs/1904.10098

[5] https://arxiv.org/abs/2203.08509
[6] http://proceedings.mlr.press/v108/bernstein20a.html
[7] https://proceedings.mlr.press/v124/squires20a.html

**Questions:**

- In think the paper would benefit from an extended experimental section, e.g., by validating against the methods described above.
- What is the additional computational cost of the method, when a possibly non-convex objective has to be optimized instead of the simpler objective by Huang?

---

> ### Author Response · Authors · 2023-11-20
>
> Dear Reviewer oMj1,
>
> Thank you very much for your efforts in reviewing this paper.
>
> **Q1. This seems too incremental as a conference contribution and in my opinion is of insufficient originality.**
>
> **A1.** The choice of the kernel is a crucial step in kernel-based methods, as it directly impacts the accuracy of the results. Unlike previous approaches, which often heuristically select kernel parameters manually, our method introduces an optimization-based approach for adaptive kernel parameter selection. Our approach dynamically considers the characteristics of the given data and selects appropriate kernel parameters accordingly. Introducing additional parameters poses a challenge for existing conditional likelihood-based methods to supervise model parameters. Therefore, we propose a supervised approach based on mutual information. Our experimental results demonstrate the stability improvement of our proposed method compared to existing approaches.
>
> **Q2. Several key references missed.**
>
> **A2.** Our proposed method introduces a scoring approach that models relationships in a more generalized manner, focusing on the local associations between given potential independent and dependent variables.
> Based on our understanding, references [2-5] predominantly
> focus on searching for DAGs under the assumption of a relatively simple local relationship model.
> While the mentioned references [6-7] appear to address extreme scenarios involving hard intervention targets or missing variables in the field of causal discovery.
> Given the extensive scope and diverse content of causal discovery, we have chosen to align our work with a narrower field relevant to our method.
>
> **Q3. What is the additional computational cost of the method, when a possibly non-convex objective has to be optimized instead of the simpler objective by Huang?**
>
> **A3.** Due to the introduction of additional parameters (parameters in the kernel applied to the dependent variable), our method has a relatively higher computational complexity compared with score function proposed by Huang et al.
> However, the primary computational bottleneck in Gaussian processes lies in the Cholesky decomposition of $K_{PA}$ with a computational complexity of $O(n^3)$.
> Therefore, the computational complexity of both our method and Huang are primarily dominated by this term.

---

> > ### Comment · Reviewer_oMj1 · 2023-11-20
> >
> > I thank the authors for responding to my questions and the clarifications.
> >
> > Regarding Q2: I understand this scope of your method and that the focus of [2-5] might be different, and this is was not a criterion for my final rating. However, readers should, in my opinion, still be made aware of recent research on score-based causal discovery and recent methods should be included in experimental evaluations (at least the score-based methods [2-4]). I agree, however, that the methods [5-7] are not of critical importance.
> >
> > I again thank the authors for clarifications, but will retain my current rating.

---

### Official Review · Reviewer_2GNL · 2023-11-01

**Soundness:** 2 fair
**Presentation:** 3 good
**Contribution:** 1 poor
**Rating:** 3
**Confidence:** 3

**Summary:**

This paper emphasizes the importance of selecting proper kernel parameters when training a score-based causal inference model on a reproducing kernel Hilbert space (RKHS). The paper illustrates two counter-examples where the existing method may fail and proposes a marginal likelihood maximization principle to maximize the log-likelihood of the joint distribution of all variables. It later empirically shows its superiority on synthetic and real datasets.

**Strengths:**

This paper highlights the fundamental problem of kernel choices in the field of kernel regression as a part of score-based causal inference.

**Weaknesses:**

1. This paper is highly related to Huang et al. 2018, while the comparison is not sufficient, which dilutes the marginal contribution of this paper. Some of the comparisons are even misleading to some extent. See the Question part.

2. This paper states the criticality of choosing the right kernel. However, it keeps the original kernel choice as Huang et al. 2018 ("More specifically, we utilize the widely-used Gaussian kernel throughout the paper" on page 5). The lack of in-depth discussions on different choices of kernels severely undermines the completeness of the discussion.

3. This paper seems to have reproduced Huang et al. 2018 incorrectly. The variance parameter of Marginal Likelihood of Huang et al. 2018 is actually trainable ("where the hyperparameter \hat{\sigma}^2_i, as the noise variance, is learned by maximizing the marginal likelihood with gradient methods", page 1555 of https://dl.acm.org/doi/pdf/10.1145/3219819.3220104), while the authors of this paper state that the width is chosen in a manual and pre-defined way in Huang et al. 2018 and reproduce it as fixed.

**Questions:**

1. I don't quite follow the first example given in the Motivation section by the authors to attack the weakness of Huang et al. 2018. The authors give an example to show the dependency between the error and the regressor under the RBF kernel, boiling it down to the inadequate model capability to capture the causal relationship. As I understand it, the linear function class on the RKHS induced by RBF kernel is capable to be consistent with any underlying model. For example, Theorem 8 of "On Consistency and Robustness Properties of Support Vector Machines for Heavy-Tailed Distributions" ensures the consistency once the RKHS is dense in the L$_1$ space, while Theorem 4.63 of "Support Vector Machines", https://pzs.dstu.dp.ua/DataMining/svm/bibl/Support_Vector.pdf proves that the RKHS of the RBF kernel is dense in L$_p$ for $p \geq 1$. Instead, an alternative explanation may be the finiteness of samples in practice. Another problem is that the term "correlated" is not equivalent to "dependent"; the error seems to be linearly uncorrelated with the regressor.

2. I also have some questions on the second example given in the Motivation section. I think if one is going to test the (conditional) independence of X_1 and X_3 given X_2, they should regress X_1 on X_2 and X_3 on X_2 and compute the correlation between the errors according to Huang et al. 2018. I wonder about the explanations of regressing X_2 on X_1 instead and the reason why it leads to a spurious edge between X_1 and X_3.

3. The algorithm proposed by the authors is to maximize the marginal log-likelihood \log p(X = x, PA = pa | f, \sigma_x), while the corresponding term in Huang et al. 2018 is \log p(X = x | PA, f, \sigma_x). What is the difference in their dynamics that leads to potential performance differences needs more discussion？

---

> ### Author Response · Authors · 2023-11-20
>
> Dear Reviewer 2GNL,
>
> Thank you very much for your efforts in reviewing this paper. We identify there exist some misunderstandings. And here are some clarification of your concerns.
>
> **Q1. Lack of in-depth discussions on different choices of kernels.**
>
> **A1.** In this paper, we do not select kernel across different kernel families. What we do in this paper is to first give a specific characteristic kernel (such as Gaussian kernel used in the paper) and apply it to the dependent variable, and then optimize the parameters of the kernel (such as the width in the Gaussian kernel) to find the optimal parameters that best fit the data.
>
> **Q2. The variance parameter of Marginal Likelihood of Huang et al. 2018 is actually trainable.**
>
> **A2.** We believe there might be some misunderstanding here. What we want to highlight is that our method can optimize the parameters of the kernel function **imposed on the dependent variable, which is pre-defined heuristically and fixed in the existing paper**.
> Specifically, in Eq.6 of our paper (i.e., $\phi(X; \sigma_x) = f(PA; \theta) + \epsilon$), the kernel function parameter $\sigma_x$ involved in $\phi(X)$ imposed on $X$ is learnable in our approach, while it is fixed in Huang et al.'s method. Due to the introduction of additional learnable parameters, we extended the regression model in RKHS and accordingly use joint distribution likelihood $p(X, PA |\\bf{\sigma})$ to simultaneously optimize $\sigma_x$ and other learnable parameters (i.e., $\theta$ in $f$) in the model including the mentioned noise variance.
>
> **Q3. The linear function class on the RKHS induced by RBF kernel is capable to be consistent with any underlying model.**
>
> **A3.**  When the kernel is characteristic, it indeed possesses the property of fitting any function. However, what we aim to compare here are the differences in the models between our approach and Huang et al.'s method.
> In general, both methods assume a relationship model, which has the form: $g(X) = f(PA_X) + \epsilon$ and $\epsilon$ is the noise.
> The difference lies in the fact that in Huang's method, the parameters in $g$ are heuristically pre-defined and fixed, resulting in the fixed features $g(x)$.
> This assumes that , for such a pre-fixed $g(x)$, it is generated by a non-linear mapping $f$ of the independent variable $x$ with its parent variables $pa$, plus the independent noise $\epsilon$. This assumption is often hard to satisfy with most manually-selected $g$ and is not related to the functional space induced by characteristic kernel.
> That's why there are still the dependency between the error and the regressor.
> In contrast, our method optimizes parameters in both $f$ and $g$ simultaneously, which can select more proper parameters of $g$ with a suitable feature space and can better estimates the independent noise.
> This is what we want to express in the Motivation part.
>
> **Q4. Why it leads to a spurious edge between $X_1$ and $X_3$ in Caese 2 of Motivation?**
>
> **A4.** The GES method searches for all possible combinations of parent variables for a target variable and identifies the combination with the highest score. In contrast, in Huang et al.'s method, the score when $PA_{X_3} = (X_1, X_2)$ is higher than when $PA_{X_3} = (X_2)$, leading to incorrect edges. We attribute this error to the inability of Huang et al.'s method to adequately capture the relationships between $X_2$ and $X_1$ as well as between $X_3$ and $X_2$. This results in the estimated noise $\widetilde{\epsilon}_3$ not being independent of $\widetilde{\epsilon}_2$, as evident in Figure 1(d).
>
> **Q5. What is the difference in their dynamics that leads to potential performance differences?**
>
> **A5.** The likelihood of the conditional distribution is insufficient to constrain the parameters of the kernel function applied to the dependent variable. From the conditional distribution, we can observe that $p(x|pa, f, \sigma_x) = p(f(pa) + \widetilde{{\mathbf \epsilon}}| pa, f, \sigma_x) = p(\widetilde{\epsilon})$, lacking the term $E[\log|\mathrm{det} J_i|]$ in Eq.11 of the paper. Consequently, it will fail to capture real relationships if we use $\log p(X = x | PA, f, \sigma_x)$ as the score function with trainable parameters in $\phi(X)$. Specifically, it will project all samples into a very small range of the feature space with excessively high scores, which is uninformative for causal discovery purposes. Therefore, we must use $\log p(X = x, PA = pa | f, \sigma_x)$ as our score function, driven by the differences in the assumed models.

---

### Official Review · Reviewer_qtuB · 2023-11-03

**Soundness:** 3 good
**Presentation:** 3 good
**Contribution:** 2 fair
**Rating:** 6
**Confidence:** 3

**Summary:**

The paper proposes a kernel-based method for causal discovery. The criterion for the dependency estimation is derived through mutual information between noise term and independent variables. The main claim is that kernel hyper-parameters can be optimized by the proposed criterion, which can be seen as the marginal likelihood of the joint distribution. The performance is verified through synthetic and benchmark function.

**Strengths:**

The hyper-parameter selection for the kernel-based causal discovery is obviously important, and the marginal likelihood based approach provides a reasonable criterion.

**Weaknesses:**

Differences from Huang et al. (2018) are somewhat weak. Some contents are similar.

**Questions:**

Although the definition of S is different, Section 4 is quite similar to Huang et al. (2018). Is there any significant technical difference compared with Huang et al. (2018) in this part?

---

> ### Author Response · Authors · 2023-11-20
>
> Dear Reviewer qtuB,
>
> Thank you very much for your efforts in reviewing this paper.
>
> **Q1. Is there any significant technical difference compared with Huang et al. (2018) in this part?**
>
> **A1.** Section 4 is to demonstrate that our proposed score exhibits the property of local consistency, ensuring that our score can collaborate with the GES method for accurate causal discovery.
> Due to differences in the assumed models between our method and Huang et al., Eq.3 in our paper represents a joint distribution $p(PA_X, X | {\pmb \sigma})$, whereas in Huang et al., it represents a conditional distribution $p(X | PA, {\pmb \sigma})$. Additionally, the score in Huang et al. proposed does not involve optimizing kernel function parameters (i.e., $\sigma_x$ in Eq.6 of the paper) imposed on $X$, whereas our method simultaneously optimizes these parameters along with other parameters in the model, selecting optimal kernel parameters for the given data.
> Therefore, our approach differs in terms of both the model and the corresponding score function from Huang et al.

---

### Meta-Review · Area_Chair_YRwM · 2023-12-07

**Metareview:**

The authors propose an extension for the score-based causal discovery (Huang et al., 2018) for automating kernel selection. (In particular, to simultaneously optimize the kernel bandwidth of Gaussian kernel in Huang et al., 2018 via maximum marginal likelihood). The authors illustrate advantages of the proposed approach over heuristic kernel selection methods.

Automating kernel selection is a crucial problem for kernel methods, the proposed approach addresses this problem for kernel regression in the scored-based causal discovery framework. Some of the reviewers give concerns about the relation with Huang et al., 2018, and comparison with other approaches on the approach of scored-based causal discovery. It is better to not only highlight the difference and advantages of the proposed method over the work of Huang et al., 2018, but also compare with other approaches on score-based causal discovery. Overall, the submission is interesting, but is not ready yet for publication. The authors may incorporate reviewers' comments to improve the work.

**Justification For Why Not Higher Score:**

+ The reviewers concern that the submission is closely related to Huang et al, 2018. However, the authors may not elaborate the difference clearly yet. Although automating kernel selection is important for Huang et al., 2018, some reviewers also concern about its limitation for kernel bandwidth selection for Gaussian kernel.

+ It is also important to not only show advantages over Huang et al., 2018, but also other approach on scored-based causal discovery (currently, I feel that it narrows the contribution by focusing on the work of Huang et al., 2018)

**Justification For Why Not Lower Score:**

N/A

---

### Decision · Program_Chairs · 2024-01-16

Reject